

# A bibliometric analysis in gene research of myocardial infarction from 2001 to 2015

Huaqiang Zhou[1,2,*], Wulin Tan[1,*], Zeting Qiu[1,2,*], Yiyan Song[2] and Shaowei Gao[1]

[1] Department of Anesthesia, The First Affiliated Hospital of Sun Yat-sen University, Guangzhou, China
[2] Zhongshan School of Medicine, Sun Yat-sen University, Guangzhou, China
[*] These authors contributed equally to this work.

## ABSTRACT

**Objectives**. We aimed to evaluate the global scientific output of gene research of myocardial infarction and explore their hotspots and frontiers from 2001 to 2015, using bibliometric methods.

**Methods**. Articles about the gene research of myocardial infarction between 2001 and 2015 were retrieved from the Web of Science Core Collection (WoSCC). We used the bibliometric method and Citespace V to analyze publication years, journals, countries, institutions, research areas, authors, research hotspots, and trends. We plotted the reference co-citation network, and we used key words to analyze the research hotspots and trends.

**Results**. We identified 1,853 publications on gene research of myocardial research from 2001 to 2015, and the annual publication number increased with time. Circulation published the highest number of articles. United States ranked highest in the countries with most publications, and the leading institute was Harvard University. Relevant publications were mainly in the field of Cardiovascular system cardiology. Keywords and references analysis indicated that gene expression, microRNA and young women were the research hotspots, whereas stem cell, chemokine, inflammation and cardiac repair were the frontiers.

**Conclusions**. We depicted gene research of myocardial infarction overall by bibliometric analysis. Mesenchymal stem cells Therapy, MSCs-derived microRNA and genetic modified MSCs are the latest research frontiers. Related studies may pioneer the future direction of this filed in next few years. Further studies are needed.

Corresponding author
Shaowei Gao,
gaoshw5@mail.sysu.edu.cn,
gscfwid@gmail.com

## INTRODUCTION

Myocardial infarction (MI), a life-threatening condition, occurs when lack of blood flow, causing the irreversible death of heart muscle. The incidence of MI has significantly increased in the past few decades, and MI remains a leading cause of death worldwide. With the development of genetic technology, gene researches have widely launched in MI. Researchers explore the genetics about MI and apply gene therapy for better treatment efficacy. Many genes have been identified an association with MI, such as PCSK9 and

TCF21 (*Braitsch et al., 2013*; *Kathiresan & Myocardial Infarction Genetics, 2008*). Genome-wide association studies also have found 27 genetic variants that are associated with an increasing risk of MI (*O'Donnell & Nabel, 2011*). Although numerous papers focused on gene research of MI, there are limited attempts to analyze them systematically.

There are several bibliometric studies focused on cardiovascular diseases research. Mark et al. performed a global bibliometric analysis from 1999 to 2008 to evaluate trends disaggregated by country (*Huffman et al., 2013*). Publications have increased substantially in the past decade, but low-income countries with higher disease burdens still have a lower output. Bloomfield and AI-Kindi confirmed this phenomenon separately by the bibliometric analysis from 52 African countries and the Middle East (*Al-Kindi et al., 2015*; *Bloomfield et al., 2015*). *Shuaib et al. (2015)* focused on the citation frequency of top 100 cited cardiovascular articles. However, a global bibliometric analysis of the gene research of MI has not yet been performed.

Bibliometric analysis is a widely used quantitative method to examine the knowledge structure and development in research fields (*Guler, Waaijer & Palmblad, 2016*). CiteSpace V is one of the bibliometric visualization tools for visualizing and analyzing emerging trends and transition patterns in scientific literature, which was developed by Chaomei Chen in 2004 (*Chen, 2004*; *Chen, 2006*; *Chen, Ibekwe-SanJuan & Hou, 2010*; *Synnestvedt, Chen & Holmes, 2005*). Now it has been widely used to evaluate the productivity of institutions, countries and authors; identify international collaborations and geographic distributions; and explore research hotspots and frontiers in specified fields (*Chen, Dubin & Kim, 2014*). In the present study, we performed a bibliometric analysis of articles on the gene research of MI indexed from 2001 to 2015, by using CiteSpace V to explore the research trend and hot spots.

## MATERIALS & METHODS

We performed online retrieval from the Web of Science Core Collection (WoSCC) of Thomson Reuters on Oct 1, 2016. We used the key words ''myocardial infarction'' and ''gene'' to retrieve research articles or reviews between 2001 and 2015. We collected the following basic info for each article: authors, title, abstract, institution, country/region, journal, keywords, and references. The search queries were listed in Table S1.

Articles or reviews that meet the following criteria were included: (1) The time span is between 2001 and 2015; (2) articles indexed in WoSCC; (3) articles on gene research of MI, including original research and reviews; (4) articles with basic info. The following document were excluded: (1) Meeting abstracts, proceedings, corrected articles, and repeated articles; (2) unpublished documents without enough information for further analysis.

The downloaded data were analyzed based on Web of Science database literature analysis report and export information function. Then we used CiteSpace V (64 bits) to analyze publication outputs and construct knowledge maps. In addition, we also used software VOSviewer (version 1.6.6) for better network visualizations in some cases (*Ozsoy & Demir, 2017*; *Van Eck & Waltman, 2010*). In this paper, the individual network was derived from

**Table 1  Ranking of countries and institutions that published articles on the gene research of myocardial infarction indexed in the Web of Science during 2001–2015.**

| Rank | Country | Counts (%) | Institution | Counts (%) |
|---|---|---|---|---|
| 1 | USA | 509(27.469) | HARVARD UNIV | 88(4.749) |
| 2 | CHINA | 297(16.028) | KAROLINSKA INST | 37(1.997) |
| 3 | JAPAN | 205(11.063) | UNIV WASHINGTON | 35(1.889) |
| 4 | GERMANY | 180(9.714) | UNIV TORONTO | 35(1.889) |
| 5 | ITALY | 128(6.908) | LEIDEN UNIV | 29(1.565) |
| 6 | ENGLAND | 102(5.505) | HUAZHONG UNIV SCI TECHNOL | 28(1.511) |
| 7 | NETHERLAND | 97(5.235) | KAROLINSKA UNIV HOSP | 27(1.457) |
| 8 | CANADA | 97(5.235) | OSAKA UNIV | 25(1.349) |
| 9 | SWEDEN | 81(4.371) | TECH UNIV MUNICH | 24(1.295) |
| 10 | FRANCE | 62(3.346) | DUKE UNIV | 23(1.241) |

the 50 most cited articles published in a one-year time slice. TFIDF weighting was used to analyze the content of each cluster (*Merigo & Montserrat, 2010*). Finally, we also applied burst detection to investigate the growth rate of citations or keywords (*Lee, Chen & Tsai, 2016*).

## RESULTS

Based on the selection criteria, 1,853 publications about gene research of MI were indexed in WOSCC from 2001 to 2015, and were included in the study. Much of publications were research articles (1,799, 97.1%), followed by review articles (54, 2.9%). English (98.4%) is the primary language used by scholars, while the remaining 1.6% of the publications were written in Russian, Spanish, German and Portuguese. The number of publications (Table S2) showed a gradually increased trend over time which rose from 82 in 2001 to 164 in 2015 and peaked in 2015.

More than 494 scholarly journals have published articles on MI research. The 10 most active journals are presented in Table S3. They published 444 articles and accounted for 24.0% of all articles included in this study. Circulation published the highest number of articles (66, 3.616%), followed by Atherosclerosis (60, 3.238%), Plos One (48, 2.590%), American Journal of Physiology Heart and Circulatory Physiology and Journal of Molecular (42, 2.267%) and Cellular Cardiology (42, 2.267%).

The 1,853 articles originated from 69 countries and territories (Table 1, Fig. 1). The top 10 countries are composed of six European countries, two Asian countries and two North American countries which accounted for 94.87% of the total number of publications. USA, China, Japan, Germany and Italy were the most productive countries.

The 1,853 articles were published by more than 500 research institutions (Table 1, Fig. 2). The top 10 institutions with the greatest outputs in this area totally published 351 articles, accounting for 18.94% of publications. The first major research echelon was led by Harvard University, followed by Karolinska Institutet, University of Washington, University of Toronto, and Leiden University.

Gene research of MI were distributed in more than 60 special research areas. Table S4 shows the top 10 research areas from 2001 to 2015. Cardiovascular System Cardiology
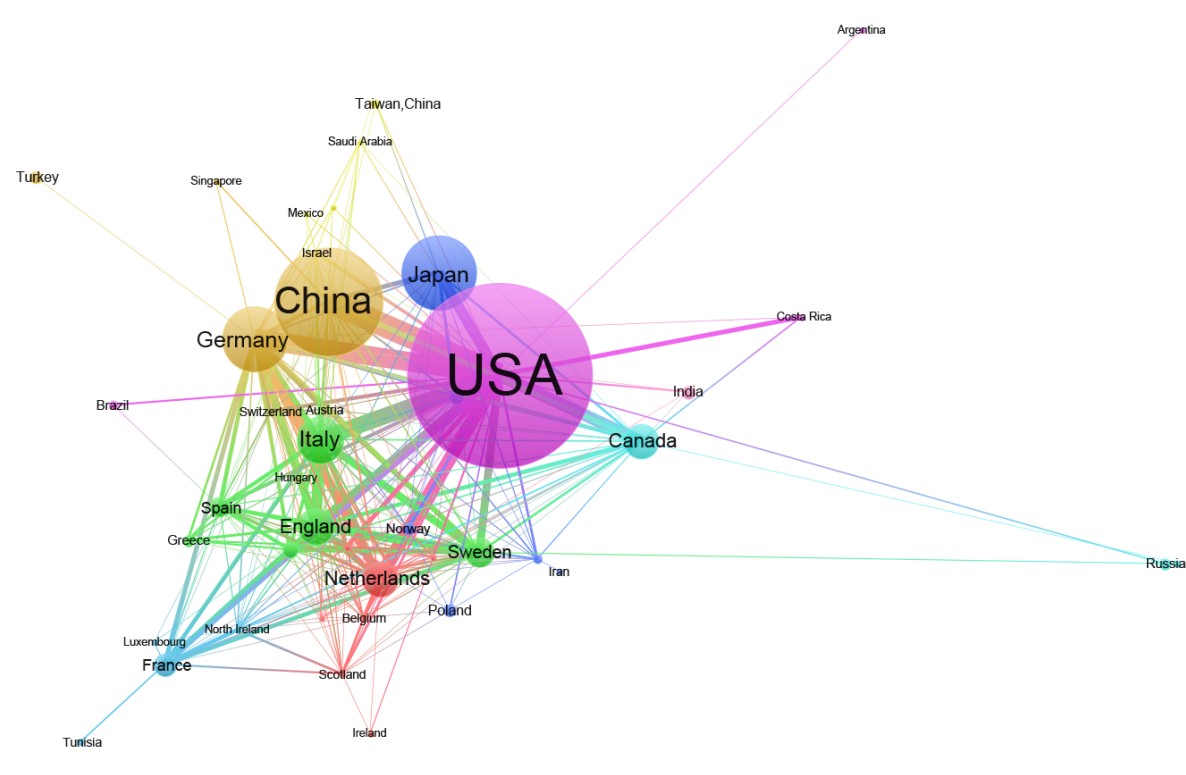

VOSviewer

**Figure 1 Map of countries and territories of groups that published articles on the gene research of myocardial infarction during 2001–2015.**

accounted for the largest number of publications (45.278%), followed by research experimental medicine (11.387%), genetics heredity (10.362%), cell biology (9.498%), and hematology (8.852%).

The 1,853 articles were written by more than 11,225 authors. The top 10 authors publishing articles were listed in Table S5. Hamsten Anders from the Karolinska Institutet, Psaty BM and Schunkert H published the most articles (25 records) and accounted for 1.349% of all published articles. We also analyzed author citations, by using Citespace V, and constructed co-citation maps to evaluate the scientific relevance of publications. As shown in Fig. 3, the largest nodes were Pfeffer MA (179 citations), Ridker PM (143 citations), Frangogiannis NG (125 citations) and Libby P (119 citations), indicating their important role in MI research. Additionally, there were seven large citation clusters in Fig. 3.

In the present study, we used CiteSpace V to construct a knowledge map of keyword co-occurrence with 147 nodes and 947 links (Fig. 4) and identified the top 20 keywords in publications from 2001 to 2015 (Table S6), according to frequency and citation counts. The top keywords were 'myocardial infarction', 'coronary artery disease', 'heart failure', 'expression', 'polymorphism', 'atherosclerosis', and 'gene expression'.

We used CiteSpace V to detect burst keywords. Burst keywords are considered as indicators of research frontiers or emerging trends over time. Figure 5 shows the top 20 keywords with the strongest citation bursts. The strongest ones include cytokine,

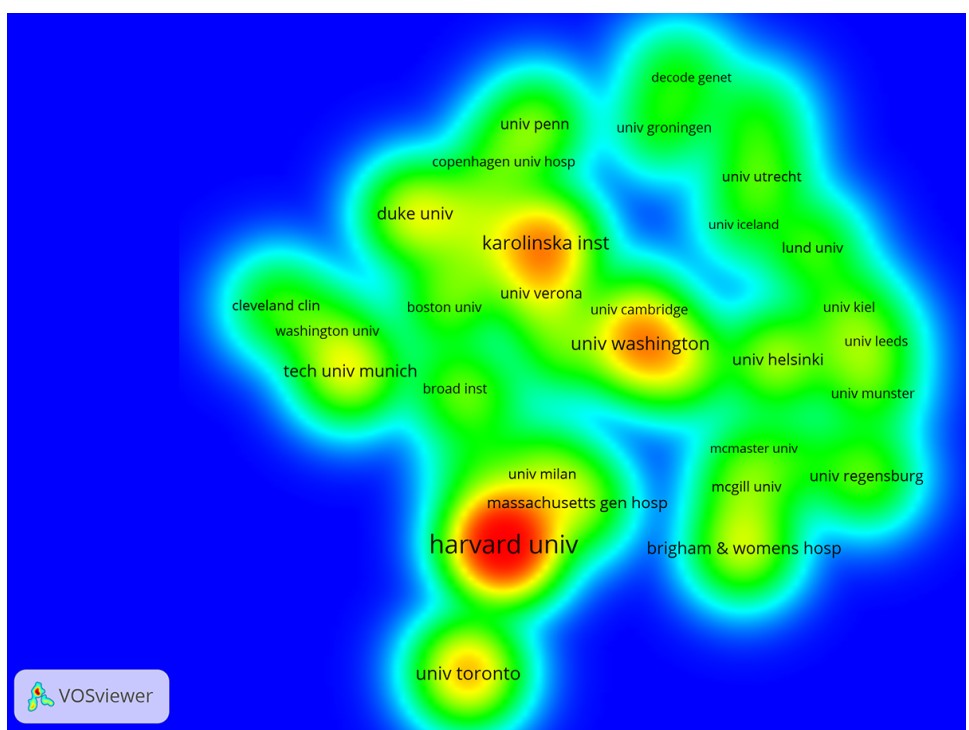

**Figure 2** Institutions that published articles on the gene research of myocardial infarction during 2001–2015.

angiotensin converting enzyme, genetics, mutation and mesenchymal stem cell. The recent burst of keywords were microRNA, mesenchymal stem cell, oxidative stress and gene therapy.

We constructed a co-cited reference map and explored changes associated with the key clusters of articles. The network contains 491 nodes and 1,374 links. The Modularity Q was 0.7863 and the Mean Silhouette was 0.3218 (Fig. 6). There were initially 71 clusters, and we filtered out small clusters with low silhouette. To learn more about the development of a cluster in a certain period, we also construct a reference co-citation time-view map (Fig. 7). According to the citation frequency and centricity, we selected hotspot references in each period of time.

## DISCUSSION

When considered with the impact of a journal, we found that the top 10 active journals all had an IF >3.0, and more than 33.3% (3/10) of the top 10 active journals had an IF >10 which account for 7% of the total number of included publications, including Circulation (IF2015 = 17.047), Journal of the American College of Cardiology (IF2015 = 17.759) and Circulation Research (IF2015 = 11.551). When compared with the high-IF rate of all MI articles (7.5%), gene research of MI was still intensively published in high-IF journals.

The United States was the leading country in gene research of MI over the past 15 years, and China was the only developing country in the leading group, demonstrating its

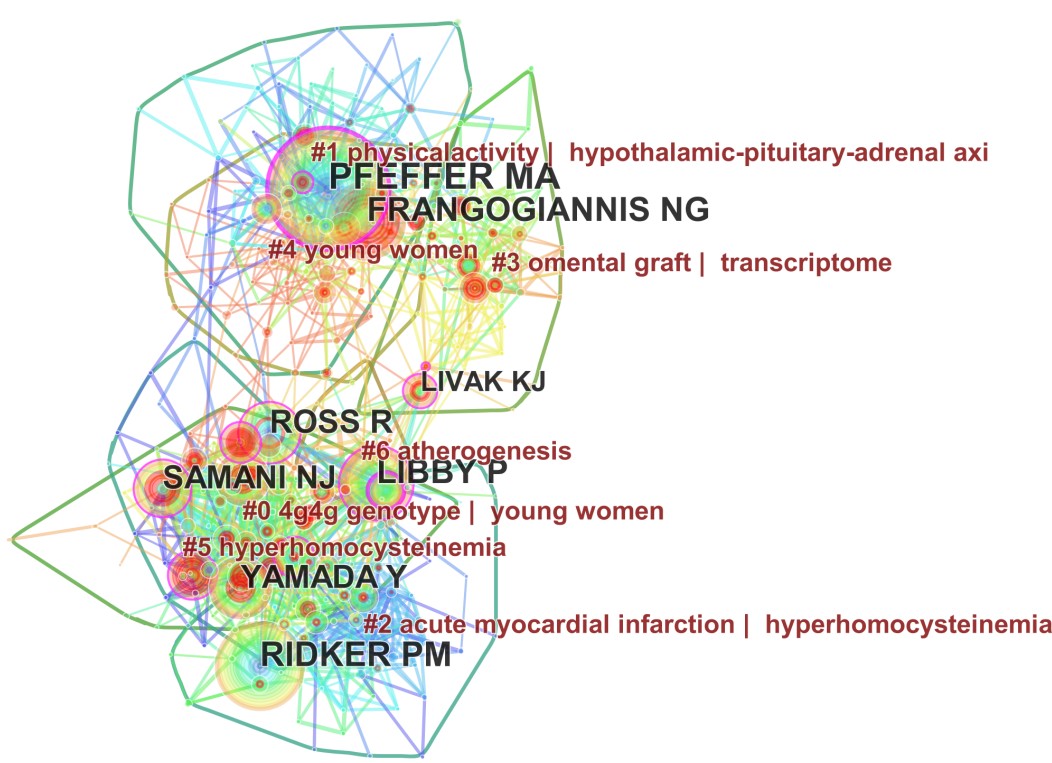

**Figure 3** Co-citation map of authors who published articles on the gene research of myocardial infarction during 2001–2015.

significant progress in the life sciences during the past 15 years. Our study is not the first investigation showing the pioneer countries in scientific output. These results were roughly consistent with previous cardiovascular disease bibliometric studies (*Huffman et al., 2013*). A 2017 study showed that the USA has the highest number of strong citation links with other countries, and China has the highest growth (*Gal, Glänzel & Sipido, 2017*). Although the number of publications from Latin America, Africa, and the Middle East increased in the past decade, these low- and middle-income countries with higher disease burdens still lag behinds developed countries, which are needed more target research investments and international collaboration (*Al-Kindi et al., 2015*; *Bloomfield et al., 2015*; *Colantonio et al., 2015*; *Jahangir, Comandé & Rubinstein, 2011*).

In Fig. 3, the color of nodes indicated how new the associated reference was. Cool colors referred to old researches, warm colors new researches (*Chen et al., 2012*). Consequently, Ridker PM (in the bottom) and Pfeffer MA (in the top) represents the basis of MI research. The former concentrates on clinical associations, such as hyperhomocysteinemia, atherosclerosis and so on (*Libby, Ridker & Hansson, 2011*; *Zee et al., 2007*). The latter hammers focus on the pathophysiology, such as changes of the renal function (*Verma et al., 2007*). In the middle of Fig. 3, warm-color nodes and linkages, leaded by Frangogiannis NG and Libby P, indicated that transcriptome was new directions (*Fiedler et al., 2011*; *Ounzain et al., 2015*). Young women label appeared twice in warm-color clusters, contacting closely

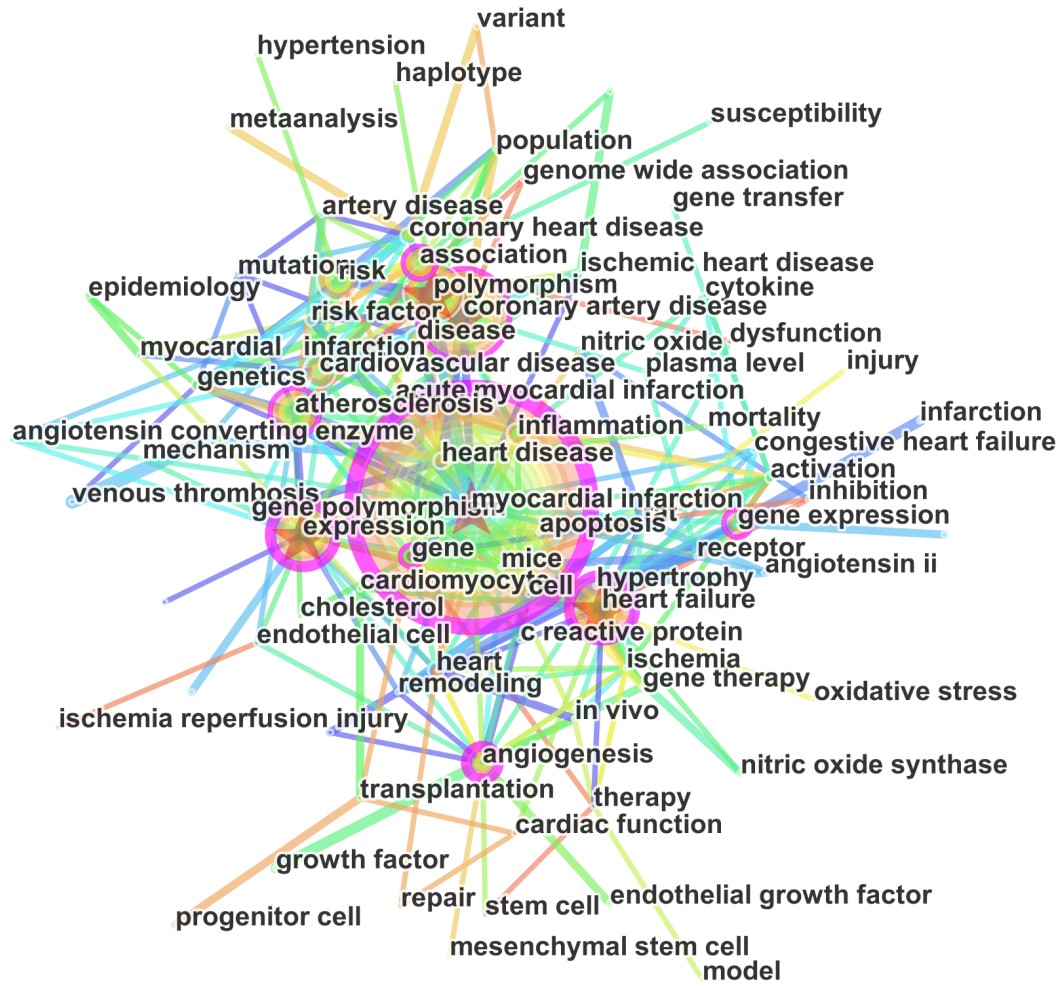

**Figure 4   Keywords networks based on articles on the gene research of myocardial infarction during 2001–2015.**

with genotype, which may have indicated young MI women may be associated with genetics variants. Some articles found that young women with acute myocardial infarction have more comorbidity and higher in-hospital mortality than young men (*Gupta et al., 2014*). Young women with MI more often have a family history of premature MI, which was addressed in a study of 398 families in which 62 vascular biology genes were evaluated. Single nucleotide polymorphisms in several thrombospondin genes were significantly associated with familial premature MI (*Topol et al., 2001*).

The topics involved in gene research of MI can been delineated in the keywords assigned to each article. Keywords provide a reasonable description of research hotspots, whereas burst words represent new research frontiers (*Chen, Dubin & Kim, 2014*). Adjacent keywords are often assigned to the same articles. For example, in Table S6, 'coronary artery disease', 'atherosclerosis', 'myocardial infarction', and 'heart failure' are closely related to coronary artery disease, and are different pathophysiology stages of CHD. Gene expression

| Keywords | Year | Strength | Begin | End | 2001-2015 |
|---|---|---|---|---|---|
| cytokine | 2001 | 13.4142 | 2002 | 2006 | |
| angiotensin converting enzyme | 2001 | 13.0615 | 2001 | 2006 | |
| genetics | 2001 | 11.665 | 2006 | 2009 | |
| mutation | 2001 | 10.4072 | 2001 | 2004 | |
| mesenchymal stem cell | 2001 | 9.8746 | 2011 | 2015 | |
| injury | 2001 | 9.8208 | 2012 | 2015 | |
| endothelial growth factor | 2001 | 9.7786 | 2008 | 2010 | |
| oxidative stress | 2001 | 9.5117 | 2012 | 2015 | |
| microrna | 2001 | 9.1851 | 2013 | 2015 | |
| haplotype | 2001 | 9.0539 | 2006 | 2009 | |
| congestive heart failure | 2001 | 9.0239 | 2003 | 2005 | |
| stem cell | 2001 | 8.908 | 2010 | 2012 | |
| cardiac function | 2001 | 8.888 | 2010 | 2015 | |
| rat | 2001 | 8.3051 | 2001 | 2005 | |
| infarction | 2001 | 8.1902 | 2001 | 2005 | |
| remodeling | 2001 | 8.1658 | 2005 | 2007 | |
| gene therapy | 2001 | 7.9672 | 2011 | 2013 | |
| nitric oxide synthase | 2001 | 7.6665 | 2006 | 2007 | |
| gene | 2001 | 7.5736 | 2001 | 2003 | |
| angiotensin ii | 2001 | 7.4706 | 2003 | 2005 | |

**Figure 5** Top 20 keywords with the strongest citation bursts on the gene research of myocardial infarction during 2001–2015.

is the synthesis process of gene product which leads to the appearance in the phenotype. There are many gene products being associated with MI, such as microRNA, SDF1 and so on (*Boon & Dimmeler, 2015*; *Bromage, Davidson & Yellon, 2014*). Burst words can be detected by CiteSpace, which represent words that are cited frequently in a period of time. As shown in Fig. 5, the top four burst keywords were as follow, mesenchymal stem cell, oxidative stress, injury, and microRNA. Combined with those keywords with red lines in Fig. 4 for further analysis, mesenchymal stem cells (MSCs) transplantation, microRNA and cardiac repair are forefronts of research in recent years. MSCs is a kind of non-hematopoietic stem cell with low immunogenicity, which has become the main cell for MI regenerative therapy (*Chou et al., 2014*). Many studies showed that MSCs transplantation therapy could have infarct-limiting and cardiac repair effects after MI by revitalizing the cardiac stem cell and revascularization (*Russo et al., 2014*; *Shafei et al., 2017*). Paracrine effect of MSCs could be mediated by extracellular vesicles, such as MSCs-derived exosomal microRNA (*Huang et al., 2015*; *Wen et al., 2012*). Genetic modified MSCs has been a therapeutic hotspot target for MI, which can promote secretion of paracrine factors to enhance the therapeutic effect (*Karpov et al., 2017*).

As is shown in Fig. 6 and Table S7, Cluster 0, 1, 4, and 5 play an important role. Cluster 0 (basic-fibroblast growth factor | gene expression profiles) was the largest while Cluster 1 (system biology/cell survival) and Cluster 4 (inflammation/young women) have the most

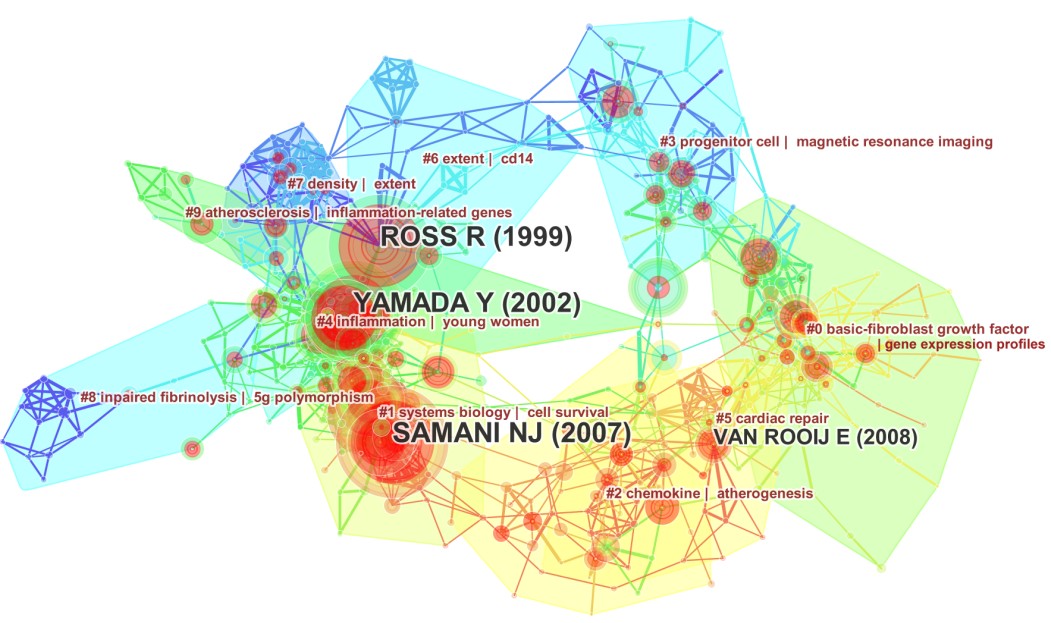

**Figure 6** Reference co-citation map of articles on the gene research of myocardial infarction during 2001–2015.

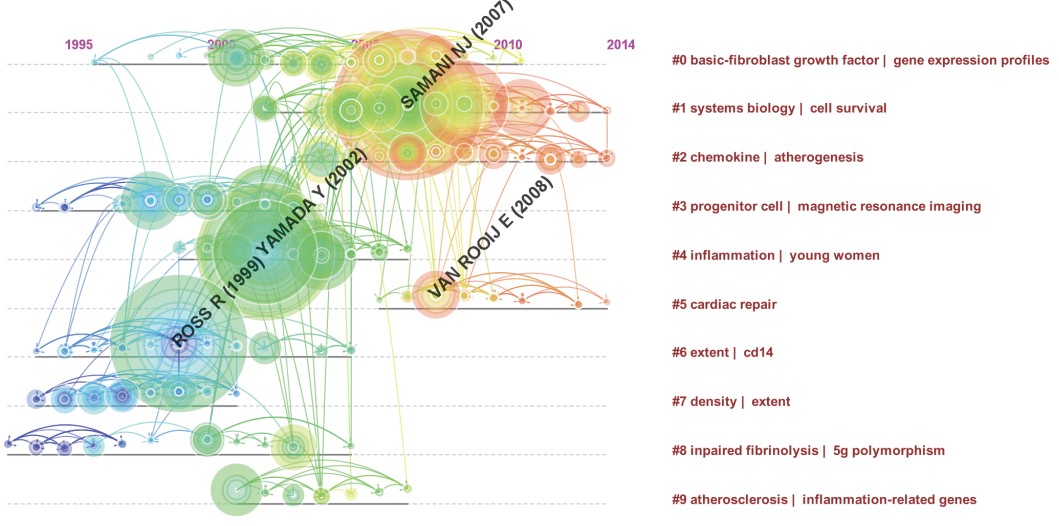

**Figure 7** Reference co-citation time-view map of articles on the gene research of myocardial infarction during 2001–2015.

burst cited-articles. The nodes and linkages of Cluster 2 (Chemokine/atherogenesis) and Cluster 5 (cardiac repair) are painted on warm color, and indicated they are the latest research, which is better visualized in Fig. 7. Besides the same warm colors, Cluster 2 and Cluster 5 have related overlapping each other, indicating relevance from aspect of literature metrology. Upregulation of chemokines is a hallmark of the inflammatory following

MI (*Cavalera & Frangogiannis, 2014*). Some researchers have suggested that chemokines may be potential therapeutic targets to promote wholesome cardiac repair in MI patients (*Cavalera & Frangogiannis, 2014*). It just so happens that ''young women'' appears once again in the Cluster 4 echoing with the previous keywords analysis, hinting at a hotspot in the gene research of MI (*Spatz et al., 2015*).

There are some limitations in our bibliometric study. The primary source of input data for CiteSpace is the WoSCC, which is more advanced at getting detailed data (e.g., journal sources, author, country and institution information). Therefore, our analysis was performed only with publications in the WoSCC without using multiple search engines (Pubmed, Ovid, Scopus and Google Scholar et al. are not included). In addition, a linguistic bias may exist because most publications in the WosCC were in English.

## CONCLUSIONS

In summary, we depicted the scientific output of gene research of myocardial infarction overall by bibliometric analysis overall. Mesenchymal stem cells therapy, MSCs-derived microRNA and genetic modified MSCs are the latest research frontiers. Related studies may pioneer the future direction of this filed in next few years. Further studies are needed.

## ACKNOWLEDGEMENTS

The authors would like to thank editors and the anonymous reviewers for their valuable comments and suggestions to improve the quality of the paper. They are also grateful to CDMG (SYSU) for assistance.

### Funding

The authors received no funding for this work.

### Competing Interests

The authors declare there are no competing interests.

### Author Contributions

- Huaqiang Zhou conceived and designed the experiments, performed the experiments, analyzed the data, contributed reagents/materials/analysis tools, wrote the paper, reviewed drafts of the paper.
- Wulin Tan performed the experiments, analyzed the data, wrote the paper, reviewed drafts of the paper.
- Zeting Qiu analyzed the data, wrote the paper, prepared figures and/or tables.
- Yiyan Song prepared figures and/or tables.
- Shaowei Gao conceived and designed the experiments, contributed reagents/materials/-analysis tools, reviewed drafts of the paper.

## Data Availability

The raw data can be directly obtained from the Web of Science Core Collection (WoSCC) of Thomson Reuters.

## Supplemental Information

Supplemental information for this article can be found online at http://dx.doi.org/10.7717/peerj.4354#supplemental-information.

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
