# Peer review of "A bibliometric analysis in gene research of myocardial infarction from 2001 to 2015"

_PeerJ, doi:10.7717/peerj.4354_

## Round 0.1 · original submission · Minor Revisions

Please follow the reviewer comments to improve the quality of your paper.

·

Basic reporting

Zhou et al aimed to evaluate the Global scientific output of gene research of MI and explore their hotspots and frontiers from 2001 to 2015, using bibliometric methods.
This is an interesting paper based on a different concept of research articles.
The use of English language is adequate throughout the manuscript.
Literature references could still be improved, by mentioning and discussing more works related to this field.
Article structure and data are satisfactorily presented.
Results and outcomes are relevant.

Experimental design

The methods are appropriate.
Search strategies have well been illustrated.

Validity of the findings

Data is sound. Conclusion is adequate. However, addition of other relevant points could improve the significance of the conclusion.

Additional comments

Even if the quality of language is adequate, due to some minor language issue, further polishing of language will improve the quality of the paper.

Reviewer 2 ·

Basic reporting

Overall the article is well-written, organized, and includes sufficient content. This review primarily suggests ways in which the authors can improve the writing and thoroughness of the paper but there are no suggested changes for methods or analysis.

1. It is suggested that in the introduction section that a paragraph be added to address other bibliometric studies that have been conducted on MI research or conducted on genetic research. This will provide the rationale for your study and indicate to the reader what this particular analysis is adding to the literature.

For example: Bibliometric Analysis of the Top 100 Cited Cardiovascular Articles ;The American Journal of Cardiology ; Volume 115, Issue 7, 1 April 2015, Pages 972-981

Cardiovascular disease research activity in the Middle East: a bibliometric analysis
S Al-Kindi, T Al-Juhaishi, F Haddad

Disparities in Cardiovascular Research Output and Citations From 52 African Countries: A Time‐Trend, Bibliometric Analysis (1999–2008)
GS Bloomfield, A Baldridge, A Agarwal

Global Cardiovascular Research Output, Citations, and Collaborations: An Ecologic, Time-Trend, Bibliometric Analysis (1999-2008)
MD Huffman, A Baldridge, GS Bloomfield… - 2013 - Am Heart Assoc

2. The article has a large number of figures and supplementary material, all of which are useful. However, the figures alone are not very informative. One option would be to see whether the figure can be expanded in order to view which label is near to which network/cluster. The other option is to show one or two figures but the remainder of data switch the figure to the supplement and the table to the paper.

Experimental design

The research question is clearly stated and the methods of mapping are highly technical.

If you refer to this website: https://sites.google.com/site/citespace101/design-rationale/how-should-i-cite-citespace

You can provide more details about the strength of this methodology and include all of the relevant citations.

Validity of the findings

The major threat to the validity of this study is the search. The authors should add a few sentences to the methods section to explain why they chose to search only in this search engine as oppose to using multiple search engines. The evidence for adequate coverage in the search strategy should be stated.

The major weakness of the paper is the discussion and conclusion. The current discussion largely focuses on a more detailed explanation of the results with some repetition. The revised discussion should highlight the findings and compare these findings with the other bibliometric studies (e.g. are they in alignment regarding productivity by region or by institution, by journal etc.?). Also, one unique component of CiteSpace is the analysis of burst keywords. The results of this analysis and particularly the relevance and implication of the identified areas should be discussed. Conclusion should be revised in order to state what exactly is the future direction of research indicated as a result of this study.

Additional comments

The remaining items are very minor

line 49: "genetic mystery" either explain what you mean by this term or remove it.
line 76: "simplily" is a typo but the term can be removed
line 96: "more likely"....I think that this statement is not supported by your evidence. Either add evidence/reference or revise it.
line 99: change American to North American so that it is clear that you are referring to the continent and not the country.

Reviewer 3 ·

Basic reporting

A well written article reviewing the literature related to myocardial infarction. According to me article is acceptable for publication after corrections are made (Minor revision)

Experimental design

Bibliyometric analysis (literature analysis) with co-citation analysis

Validity of the findings

The validity of the findings was examined by using the documents the authors uploaded in the supplementary materials. The findings are valid.

Additional comments

Analyzed in detail and given important findings on the subject but the following should be taken into consideration.

1) Authors should write a limitation of the study because the analysis were performed only with publications in the wos index (pubmed scopus et al. are not included).

2) Maps do not look nice and difficult for readers to understand. So this study was carried with the VOSviewer could have been better. Authors should read and cite the following study for corrections;

“Ozsoy, Z. & Demir, E. The Evolution of Bariatric Surgery Publications and Global Productivity: A Bibliometric Analysis, OBES SURG (2017). https://doi.org/10.1007/s11695-017-2982-1

3) If descriptive footnotes are added to the figures, readers can understand more easily

---

## Round 0.2 · accepted · Accept

Your paper is now ready for publication.